# Network Approach to Evaluate the Effect of Diet on Stroke or Myocardial Infarction Using Gaussian Graphical Model

**DOI:** 10.3390/nu17101605

**Published:** 2025-05-08

**Authors:** Jaca Maison Lailo, Jiae Shin, Giulia Menichetti, Sang-Ah Lee

**Affiliations:** 1Interdisciplinary Graduate Program in Medical Bigdata Convergence, College of Medicine, Kangwon National University, Chuncheon 24341, Republic of Korea; lailojacamaison@gmail.com (J.M.L.); jiaejshin@gmail.com (J.S.); 2Channing Division of Network Medicine, Department of Medicine, Brigham and Women’s Hospital, Harvard Medical School, Boston, MA 02114, USA; giulia.menichetti@channing.harvard.edu; 3Network Science Institute, Northeastern University, Boston, MA 02115, USA; 4Harvard Data Science Initiative, Harvard University, Boston, MA 02134, USA; 5Department of Preventive Medicine, College of Medicine, Kangwon National University, Chuncheon 24341, Republic of Korea

**Keywords:** Gaussian graphical model, dietary patterns, stroke, myocardial infarction

## Abstract

**Background/Objectives/Methods:** Current research on the link between diet and stroke or myocardial infarction primarily focuses on individual food items. However, people’s eating habits involve complex combinations of various foods. By employing an innovative approach known as the Gaussian graphical model to identify dietary patterns along with the Cox proportional model, the study aimed to identify dietary networks and explore their relationship with the incidence of stroke and/or myocardial infarction in the Korean population. The research utilized data from 84,729 participants in the Korean Genome and Epidemiological Study (KoGES), including the HEXA cohort (61,140 participants), CAVAS cohort (15,419 participants), and Ansan-Ansung cohort (8170 participants). **Results:** The network identified five dietary patterns or communities consisting of different food groups, while nine food groups did not belong to any community. The High-Protein and Green Tea Community consistently reduced the risk of stroke and myocardial infarction (MI), particularly among females. In most communities, no significant associations with stroke risk were noted in males, and the Rice and High-Calorie Beverages Community was linked to an increased risk of MI in both the total population and females. **Conclusions:** Dietary patterns derived from network analysis revealed distinct dietary habits in the Korean population, offering new insights into the relationship between diet and the risk of stroke and MI.

## 1. Introduction

In South Korea, 232 per 100,000 adults experience stroke each year, while the incidence of myocardial infarction (MI) hospitalizations stands at 43.2 cases per 100,000 population [1,2]. Globally, 85% of the 17.9 million cardiovascular (CVD) deaths are attributed to stroke and MI [3]. Given these statistics, raising awareness of the distinct signs and factors influencing survival is a crucial public health strategy

Multiple studies and systematic reviews have evaluated the effect of diet on stroke or myocardial infarction (MI), particularly focusing on the effects of individual food intake. Protective foods include whole grains (2% reduction in stroke risk and 25% reduction in MI risk) [4,5], low-fat dairy (7% reduction in stroke risk) [6,7], fruits and vegetables [8,9], fish rich in omega-3 fatty acids (6% reduction in stroke risk and 27% reduction in MI risk) [10,11], nuts (12% reduction in stroke risk) [12], and green tea (13% reduction in stroke risk) [13]. Conversely, processed meats and sugary beverages increase the risk, with sugary beverages raising stroke risk by 13% and MI risk by 22% [14,15]. Other foods, such as unprocessed red meat, white meat [16,17], and legumes [18], have been shown to either reduce or have unclear effects on stroke and MI risks.

Collectively, these findings highlight the importance of examining individual food effects; however, the human diet typically involves a combination of food groups, making it essential to investigate dietary patterns. For example, the Mediterranean Diet, characterized by a high intake of healthy oils and vegetables; a moderate intake of fish and other meat, dairy products, and red wine; and a low intake of eggs and sweets [19,20,21], has been found to reduce the risk of stroke by 12% and MI by 26% [22]. Similarly, previous studies have identified three distinct dietary patterns: (1) a prudent pattern, characterized by a higher intake of fruits, vegetables, and whole grains, which reduces stroke risk by 22%, (2) a Western dietary pattern, defined by higher intakes of red and processed meats, refined carbohydrates, and added sugars, associated with a 58% increase in stroke risk [20,21,23], and (3) an oriental pattern, characterized by a high intake of tofu, soy, and other sauces, albeit no association with MI risk [24,25]. However, the relevance of these patterns can vary across populations [20] such as in Korea, where diet patterns include a high intake of white rice, beans, vegetables, kimchi, and seaweed [26]. This variation emphasizes the need for population-specific dietary analysis, as dietary patterns offer more insight into disease associations than individual foods alone [25].

Dietary patterns in diet–disease relationships are frequently obtained using multivariate techniques such as principal component analysis, factor analysis, and cluster analysis. However, these methods have notable limitations, including a lack of a probabilistic structure, complex interpretability, and the use of deterministic groupings [27,28]. A promising alternative is the Gaussian graphical model (GGM) [29], which captures direct relationships between food groups through correlations. The GGM enhances interpretability by visualizing the network structure, identifying sparse and meaningful communities, and better accommodating the complex, high-dimensional nature of dietary data. The GGM offers greater flexibility and a more accurate reflection of the underlying dietary structure compared to traditional methods by modeling conditional dependencies within a probabilistic framework [30], making it an ideal tool for this study in understanding the complex interactions among food groups. The objectives of this research were to apply the GGM to identify dietary networks and to examine the associations of these networks with stroke or MI risk in the Korean population.

## 2. Materials and Methods

### 2.1. Study Population

This study included three cohorts from the Korean Genome and Epidemiological Study (KoGES): the Health Examinee (HEXA) cohort, the Cardiovascular Disease Association Study (CAVAS) cohort, and the Ansan-Ansung (AA) cohort. The HEXA cohort (2004–2013; 173,195 participants), the CAVAS cohort (2005–2011; 28,338 participants), and the AA cohort (2001–2002; 10,030 participants) comprised males and females aged 40 years or older from both urban and rural locations. Baseline characteristics included demographic variables, lifestyle factors, anthropometric measurements, clinical measurements, and biospecimen data. Trained evaluators conducted one follow-up survey for HEXA, four follow-up surveys for CAVAS, and nine follow-up surveys for AA, starting in 2003. Informed consent was provided by all participants. Detailed information on the three cohorts has been previously discussed elsewhere [31,32]. This study was provided with biomedical and research resources containing genetic and health information from CODA (Clinical & Omics Data Archive), the Agency for Disease Control and Prevention, Republic of Korea (CODA_S2500002-01).

The HEXA, CAVAS, and AA cohorts combined consist of 211,562 participants at baseline. Participants from invalid recruitment sites (31,306), those with implausible energy levels (5876)—for males, fewer than 800 kilocalories (kcal) and more than 4000 kcal, and for females, fewer than 500 kcal and more than 3500 kcal—participants without at least one follow-up data point (85,616), and those with self-reported diagnoses or treatment of stroke or myocardial infarction at baseline (3821) were excluded from the study. Consequently, a total of 84,729 participants from the three cohorts comprise the study population (Figure 1).

### 2.2. Outcome Definition

The main outcomes of the study were determined through self-reported responses to the following questions: (1) “Diagnosed with stroke?” and (2) “Diagnosed with angina/MI?” Both questions required “Yes” or “No” responses. At baseline, participants who answered “Yes” to either question were removed from the analysis, as shown in Figure 1.

During the follow-up period, which consisted of one follow-up period for HEXA, four follow-up periods for CAVAS, and nine follow-up periods for the AA cohort, outcomes were identified. If a participant self-reported a stroke or MI during a given follow-up period, their case was not carried forward to subsequent follow-up phases. Participants who reported both stroke and MI during the same follow-up period were classified as “self-reporting both stroke and MI”. For participants who did not self-report stroke or MI during one follow-up period, responses to the same questions in the subsequent follow-up periods were assessed to determine outcomes. Participants who consistently answered “No” to either question across all follow-up periods were classified as non-cases. Individuals with missing responses to these questions across all follow-up periods were excluded from the study.

As illustrated in Figure 1, a total of 3097 cases of either stroke or MI, 1096 cases of stroke, and 2057 cases of MI were defined in the study. Detailed identification of the outcomes for each follow-up period for all cohorts is shown in Appendix A and for each cohort is shown in Appendix A for HEXA, Appendix A for CAVAS, and Appendix A for AA cohorts.

### 2.3. Dietary Assessment

Dietary intake was evaluated with a semi-quantitative food frequency questionnaire (FFQ), which assessed participants’ consumption of 106 food items for the HEXA and CAVAS cohorts and 103 food items for the AA cohort. The FFQ was validated against 12-day dietary records showing random measurement error-corrected correlation coefficients ranging from 0.23 to 0.64. Additionally, reproducibility between two FFQs measured one year apart showed correlation coefficients ranging from 0.24 to 0.58, supporting reasonable stability over time [33].

Nine distinct food consumption categories were generated to assess dietary intake. Additionally, three portion size categories were produced to further characterize the participants’ dietary patterns [31,32,33]. The aligned 106 and 103 items were initially classified into 37 food groups based on prior research [34]. The classification was further expanded to 45 food groups based on individual food group associations between stroke or MI [8,20,21,25,26,34,35,36] and expert input from a qualified nutritionist. The composition table can be found in Appendix A.

### 2.4. Covariates

This study included baseline demographics and lifestyle variables such as age, education level, income, size of the city, marriage status, body mass index (BMI), waist circumference, history of hypertension, hyperlipidemia, diabetes mellitus, type of job, smoking status, drinking status, and exercise. The categories for each variable can be seen in Table 1.

### 2.5. Network Analysis

GGMs utilize probabilistic graphs to examine and illustrate dependency structures through a graph depicting conditional independence among variables. It requires multivariate normality and the conditional dependency of nodes [30]. These graphs consist of nodes representing variables and edges representing conditional dependency relationships. A missing edge between two nodes implies conditional independence between them given all other variables. Model selection in GGMs leads to a sparse graph representing the underlying variable patterns.

The theoretical foundation of GGM begins with a data matrix X containing n observations and *p* variables derived from a *p*-variate normal distribution. As most dietary data have a skewed distribution, the data were transformed to log_10_(1 + (grams per day)) to improve normality. Log transformation is particularly effective for handling right-skewed distributions, as it helps normalize the data and make them more suitable for statistical analysis. Additionally, it maintains interpretability across food groups, unlike alternative methods such as bestNormalize [37], which may apply different transformations to each variable. After transformation, about 15 food groups were approximated to normality.

From the inverse of the covariance matrix (also known as precision matrix), the conditional distribution of any two variables given other variables can be derived. GGMs rely on identifying conditional independence in the precision matrix, represented as an undirected graph [29,30,38]. Graphical Lasso is a regularization technique employed in GGMs to induce sparsity in the precision matrix, thus revealing the meaningful underlying structure of the data by penalizing off-diagonal elements.

The optimum tuning parameter lambda was assessed using the Stability Approach Regularization Selection (StARS) implemented in the “huge” R package [39], which selects the optimum lambda. StARS is particularly suited for high-dimensional data, where controlling the stability and reproducibility of the network structure is prioritized [30,40]. The final selected value of lambda was 0.2670. The lambda parameter was used to estimate the sparse precision matrix using the R package “glasso”. The network was visualized using the R package “igraph”, and the same package was utilized to identify network communities and the eigenvector centrality score. Community detection was performed using Multi-Level Modularity Optimization (cluster_louvain in igraph), which ranked highest in previous evaluations of igraph clustering methods. This algorithm identifies communities by maximizing modularity, grouping nodes with dense intra-community connections [41].

A community network-based score was also computed by multiplying the eigenvector score for each node by the actual daily food group intake per individual per community. Quintiles of each community network-based scores (Q1, Q2, Q3, Q4, and Q5) were also obtained for further analysis.

### 2.6. Statistical Analysis

Missing data were handled using different imputation strategies depending on the extent of the missing values. For variables with less than 5% missing data, missing values were imputed using the median value, stratified by sex to account for potential sex-specific differences. For variables with missing data greater than 5%, a separate category for missing data was created to ensure that participants with missing information were still included in the analysis. The number and percentage of participants for each covariate and its association with the risk of self-reported stroke or myocardial infarction (MI) was obtained using a Cox proportional hazards regression model. Significant hazard ratios (HRs) were used to identify covariates that can be included in the model. To assess the association of the dietary network and stroke or myocardial infarction, continuous community network-based scores and quintiles of community network-based scores were adjusted for demographics and lifestyle variables using three Cox proportional regression models. Model 1 was adjusted for age and sex (for total respondents only); Model 2 included additional adjustments for education, city, marital status, income group, job group, previous history of hypertension, hyperlipidemia, or diabetes, BMI, and waist circumference, in addition to the adjustments in Model 1; and Model 3 further adjusted for drinking status, smoking status, total energy intake, and exercise, building upon Model 2. The three models were stratified by sex, and to examine the potential effect modification of sex, interaction terms between sex and dietary networks were included as additional analysis for Model 3. Person-years were calculated by determining the number of years between the baseline survey date and the follow-up survey date where an outcome was defined. For participants without an outcome, the last follow-up survey date was used as the end date. Statistical significance was defined as a *p*-value < 0.05 unless otherwise specified. Statistical analyses were conducted using R version 4.4.0 within the RStudio environment 2023.06.21 Build 524.

## 3. Results

Table 1 summarizes demographics and lifestyle factors by total population and by sex. The median (IQR) age of the participants was 54 (48–61) years. Participants contributed a total of 508,095 person-years (median: 4.5 person-years) during which 3097 incident cases of stroke or MI were observed (stroke: 1096 cases; MI: 2057 cases) (Figure 1). Stroke or MI risk was higher for participants with lower educational attainment (≤middle school), lower income (<KRW 4 million), rural residency, single or unmarried, higher body mass index (BMI), above-normal waist circumference, smoking (both past and current), and a history of hypertension, hyperlipidemia, or diabetes, as well as occupations involving labor and agriculture or unemployment and housekeeping. Conversely, the risk of stroke or MI was lower among those who engaged in regular exercise.

The network derived from the total population revealed five communities (Figure 2). The High-Calorie and Processed Food Community (HCPF) consisted of eggs, processed soybean products, bread, blue fish, chocolates and sweets, pizza, hamburgers, and white fish, with milk as the central node. The Fruits and Dairy Community (FD) included white fruits and dairy products. Yellow and orange fruits, purple fruits, red and pink fruits, and red vegetables served as the central nodes of this community. The High-Protein and Green Tea Community (HPGT) was made up of white meat, sushi, meat soup, noodles, and green tea, with unprocessed red meat as the central node. The Rice and High-Calorie Beverages Community (RHCB) comprised soda, other drinks, rice cakes, and mixed rice, with white rice as the central node. Finally, the Vegetables Community (VEG) includes mushrooms, various vegetables, kimchi, carrots, brown vegetables, cruciferous vegetables, radishes, allium vegetables, green vegetables, potatoes, and starch, with pumpkin as the central node. Central nodes and eigenvector centrality scores for each community are summarized in Appendix A. Median network scores for each community are summarized in Appendix A.

Additionally, Figure 2 showed nine food groups not belonging to any community: cereal, crustacean, legumes, mollusks, coffee, seaweed, processed red meat, processed seafood, and nuts. These nodes were isolated during glasso regularization, resulting in a sparse network.

Table 2 presents the association between dietary network and stroke or myocardial infarction risk. High intake of foods in the HPGT community was associated with a 2% reduction in the risk of stroke or MI (HR = 0.98, *p* = 0.028). A significant interaction effect between HPGT and sex was also observed for stroke or MI risk (*p* = 0.026). In sex-stratified analyses, although not statistically significant, both males (HR = 0.97, *p* = 0.050) and females (HR = 0.99, *p* = 0.459) showed a reduction in MI risk. Similarly, FD also decreased stroke or MI risk in Q4 compared to Q1 (HR = 0.84) in sex-stratified analysis, particularly in males.

The association between dietary networks and stroke risk only can be found in Table 3. The analysis revealed that for all communities (HCPF, FD, HPGT, RHCB, and VEG), confidence intervals included 1.00, suggesting that these communities were not significantly associated with stroke risk, particularly in males. However, in females, stroke risk decreased with the intake of HPGT, specifically in Q3 (HR: 0.73) in comparison with Q1. In the total population, the risk of stroke also decreased in HCPF and VEG communities, but this was only observed when adjusted for age and sex.

Table 4 reveals the association between dietary networks and MI risk. RHCB increased the risk of MI (HR: 1.22) when Q3 was compared with Q1 in the total population. This was also evident in females, where MI risk increased in Q2 (HR: 1.24), Q3 (HR: 1.23), and Q4 (1.28) in comparison with Q1. The VEG community also exhibited increased risk in MI specifically for females (Q2 HR: 1.22; Q3 HR: 1.23). Conversely, HPGT in the total population decreased the risk of MI when Q3 (0.84) was compared to Q1. A significant interaction effect between HPGT and sex was also revealed for **MI risk** (*p* = 0.033). Interestingly, for all communities, males did not show any significant associations with MI risk in sex-stratified analysis.

When examining individual food groups outside the network, stroke or MI risk increased with the consumption of cereals and coffee, whereas a decrease in risk was exhibited with the consumption of mollusks (Figure 3). Detailed information for each food group can be found in Appendix A. While many food groups showed no significant interaction with sex, some food groups, such as legumes, nuts, and seafood mollusks, exhibit sex-dependent effects.

## 4. Discussion

This study explored the association between dietary networks and their effect on stroke or myocardial infarction (MI) risk in an 84,729-participant Korean Genome and Epidemiological Study cohort, with 3097 incident cases (1096 stroke cases and 2057 MI cases) observed over 508,905 person-years. Using a Gaussian graphical model, five distinct dietary network communities were identified: High-Calorie and Processed Food (HCPF), Fruits and Dairy (FD), High-Protein and Green Tea (HPGT), Rice and High-Calorie Beverages (RHCB), and Vegetables (VEG). Two communities demonstrated a protective effect: HPGT and FD. Two communities showed potential harmful effects: RHCB and VEG. Finally, one community had no significant effect on stroke or MI risk: HCPF.

HPGT, comprising red meat, meat soup, white meat, noodles, sushi, and green tea, was associated with a decreased risk of both stroke and MI, particularly in higher consumption quintiles, potentially due to a combination of muscle-building and antioxidant mechanisms. Green tea, a key component of HPGT, is rich in flavonoids like catechins, which have been shown to reduce cholesterol levels, prevent atherosclerosis, and enhance nitric oxide availability, thereby improving endothelial function. Human studies suggest that regular consumption of green and black tea can improve cardiometabolic risk factors, especially by enhancing endothelial function, lowering LDL cholesterol, and reducing blood pressure [13,25,42]. Meanwhile, the association between unprocessed red meat consumption and stroke risk remains inconclusive; one meta-analysis by Kim et al. reported that high consumption of red meat was associated with an increased risk of stroke [17] but another meta-analysis report did not find a significant association [16]. Notably, additional analyses of HPGT suggested that the consumption of unprocessed red meat had a protective effect against stroke or myocardial infarction, in contrast to Bernstein’s report, which stated that red meat consumption may increase stroke risk. Alternatively, Bernstein et al. suggested that a high-protein diet and alternative protein sources might offer protective benefits by lowering blood pressure and improve blood lipids [43]. In Asians, where carbohydrates like rice are staple foods, high protein intake is particularly important via a source of high-quality protein, unlike in Western countries, where meat is more commonly a dietary staple. Moreover, the cooking methods and total meat consumption in Korea may differ significantly from those in Western countries, potentially influencing the observed associations. Therefore, further research is needed to understand the effects of red meat on Asians, including Koreans. The significant interactions of HPGT and sex also suggest that high-protein foods and green tea may have different effects in men and women, possibly due to biological differences in metabolism and food intake [42,43].

FD significantly reduced the risk of stroke or MI, particularly in males. Fruits, a majority component of this community, also reduce oxidative stress and inflammation. This aligns with a systematic review by Sherzai et al., who reported that fruit consumption significantly lowers stroke risk [20]. Further evidence from Oude Griep et al. emphasized that fruit color may also influence protection to cardiovascular risk, suggesting the potential influence of specific phytochemicals [9]. In addition, dairy offers calcium, potassium, and magnesium, which support blood pressure regulation, contributing to stroke and MI prevention [6,7]. The stronger association observed in males may stem from higher baseline oxidative stress and greater susceptibility to hypertension-related complications [44], amplifying the protective effects of dietary fiber, antioxidants, and micronutrients found in this community. Further research is warranted to explore the biological and behavioral mechanisms underlying these gender-specific effects.

Conversely, RHCB was linked to an increased risk of MI, particularly in high-consumption quintiles and in females. Sweetened beverages raise blood glucose, insulin, triglycerides, and inflammatory markers, contributing to metabolic syndrome and cardiovascular risk [15,45,46], with studies linking high-sugar sweetened beverage consumption to 19% higher MI risk (RR 1.19) [15]. White rice, the central node of RHCB, is also dietary staple in Korea [26] and may have a greater glycemic part in populations with high consumption, contributing to MI precursors including diabetes and hypertension [47]. RHCB’s heightened risk in females may be attributed to lower insulin sensitivity, hormonal changes like reduced estrogen post-menopause, and dietary preferences for sweetened beverages. Supporting evidence from Hu et al. highlights increased diabetes risk, a risk factor of stroke or MI, from high white rice consumption in East Asian females [48].

The VEG community demonstrated an unexpected inverse relationship with MI, specifically for females, which opposes the protective results from previous studies [8,20,49]. Studies have shown that females tend to perceive vegetables and fruits as healthier, leading to higher consumption [50]. However, while VEG did not affect stroke risk significantly, further analysis of individual food items revealed that allium, brown vegetables, green vegetables, and carrots increased the risk of stroke, whereas kimchi increased the risk of MI. A further analysis of its sodium content was performed to examine the hypothesis that sodium elevates the risk of stroke or MI, given the link between sodium intake and cardiovascular risks [51,52,53]. Our analysis confirmed that the sodium content in allium, brown, and green vegetables, as well as carrots, was associated with an increased risk of stroke, while the sodium content in kimchi was linked to a higher risk of MI. Another explanation for this result could lie in the traditional Korean recipes that use these vegetable ingredients. For example, the main ingredients of Kimchi are green leaves, but the sodium consumed from Kimchi might have elevated the risk of MI. Likewise, brown and green vegetables and carrots are used as ingredients of other sodium-rich dishes commonly consumed among Koreans, such as jajangmyeon (black bean noodles: 947.2 mg/dish), jjamppong (spicy seafood noodles: 605.7 mg/dish), yukgaejang (spicy beef soup: 762.7 mg/portion), and other stews (257.6 mg/100 g dish) [54,55,56]. Thus, the consumption of these vegetables often occurred with dishes with a high sodium content, which could negate the protective effects of consuming vegetables exclusively.

Finally, HCPF showed no significant associations across stroke or MI, even when stratified by sex. To explore potential effects of specific food components within this community, we conducted an additional analysis by adjusting for highly correlated items (breads, chocolates/sweets, and pizza/hamburgers, Figure 2), which have been shown in previous studies to affect the risk of stroke and MI [20,25,42]. The remaining components, including eggs, soybean-based foods, milk, and blue and white fish, were analyzed to see if these might reveal clearer associations. Our findings showed a protective effect on stroke risk in males, but no significant results for females. Given that modifying the community by removing components could disrupt the relationships captured by the GGM [29], further exploration using an improved GGM approach that better accounts for complex dietary interactions and sex-specific effects is warranted to reveal potential hidden associations in the HCPF community.

Individual food items, while not belonging to a specific community in the network, also individually demonstrated associations with stroke or MI. High coffee intake was linked to increased stroke risk, especially among males, due to caffeine’s effects on blood pressure, although added ingredients such as sugar or cream or the preparation type could have elevated these risks [42]. On the other hand, moderate nut consumption was protective and can be explained by healthy fats, fiber, and antioxidant properties, which collectively benefit cardiovascular health, although they may also induce weight gain [12,18]. Similarly, mollusks, with their high omega-3 content [57]—a chemical class known for lipid regulation mechanisms [10]—were linked to a lower risk of cardiovascular events. Higher consumption of cereals revealed different associations with both stroke and myocardial infarction (MI). Although previous studies showed inconclusive and varying results depending on the type of grain [58,59,60], this study indicated a connection to an increased risk of stroke and MI.

The study presents several strengths that enhance its contribution to understanding the relationship between dietary networks and stroke or myocardial infarction. First, it employs a novel approach (GGM) to interpret the effects of dietary intake on disease occurrence by considering the complex interactions among various foods [29]. The community detection capabilities of the GGM also enabled the identification of five distinct dietary communities—HPGT, FD, HCPF, RHCB, and VEG—each with different associations with stroke and MI risks. Second, utilizing the large, well-characterized KoGES cohort facilitated more robust inferences about long-term dietary effects. Third, the analysis included a wide range of covariates, such as demographics, lifestyle, and anthropometric measures, which allowed for estimating dietary effects while minimizing confounding biases. Additionally, exploring sex-specific effects offered valuable insights into how dietary patterns differentially influence cardiovascular risks in males and females, revealing significant sex-specific associations.

However, several limitations should be noted. First, the self-reported stroke and MI diagnosis may introduce recall bias and affect the accuracy of estimates. According to a validation study for stroke and MI cases in KoGES HEXA, the positive predictive value (PPV) of self-reported disease history was 51.4% for stroke and 32.6% for myocardial infarction (MI), highlighting the need for careful consideration due to lower-than-expected validity [61]. Inaccuracies in self-reported disease history may lead to misclassification bias, potentially weakening the observed associations. This type of bias typically results in underestimating true effects rather than producing false-positive findings. Another limitation of this cohort design is the unequal distribution of follow-ups across studies, with only one follow-up in HEXA, four in CAVAS, and nine in Ansan-Ansung [31], which could introduce bias in the calculation of person-years due to the differing frequency of follow-ups. Among the 84,729 study participants, 61,140 had one follow-up dietary assessment, primarily measured at baseline [62,63]. Thus, the potential reverse causation cannot be completely excluded. Another limitation is the applicability of these findings to populations outside Korea, as dietary patterns and typical food consumption vary significantly across cultures [20]. For instance, some Korean dietary staples, such as kimchi and white rice [26], may have different health implications in other populations, emphasizing the need for population-specific dietary analyses.

Additionally, the study adjusts for histories of hypertension, diabetes, and hyperlipidemia, as these conditions are known to independently affect stroke and MI risk. However, it does not account for medication use (e.g., statins and antihypertensives), which can modify this risk [64], nor does it exclude individuals with preclinical atherosclerosis, prior cardiovascular procedures, or high *C*-reactive protein levels, all of which can influence stroke or MI [65]. Missing data were prevalent for these variables in the dataset used for this study. Future research with more comprehensive data should consider these variables to provide more precise estimates.

Finally, the assumption of multivariate normality in the dietary data, despite applying a log transformation, have not been fully satisfied, potentially affecting the accuracy of conditional dependency estimation in the GGM [38]. To better address non-normality, future studies could consider regrouping food groups to reduce zero inflation and skewness or applying nonparametric network estimation methods that are less sensitive to distributional assumptions.

## 5. Conclusions

The study underscores the complex relationships between diet and cardiovascular health, suggesting that high-protein and fruit-rich diets may offer protective effects, while high-calorie, rice-dominant diets, excessive coffee, and processed red meat intake could increase stroke or MI risks. These findings underscore the strength of the GGM in capturing interdependence among food items and their variable effects across populations, suggesting the potential of personalized dietary recommendations based on comprehensive pattern analysis in the Korean population.

## Figures and Tables

**Figure 1 nutrients-17-01605-f001:**
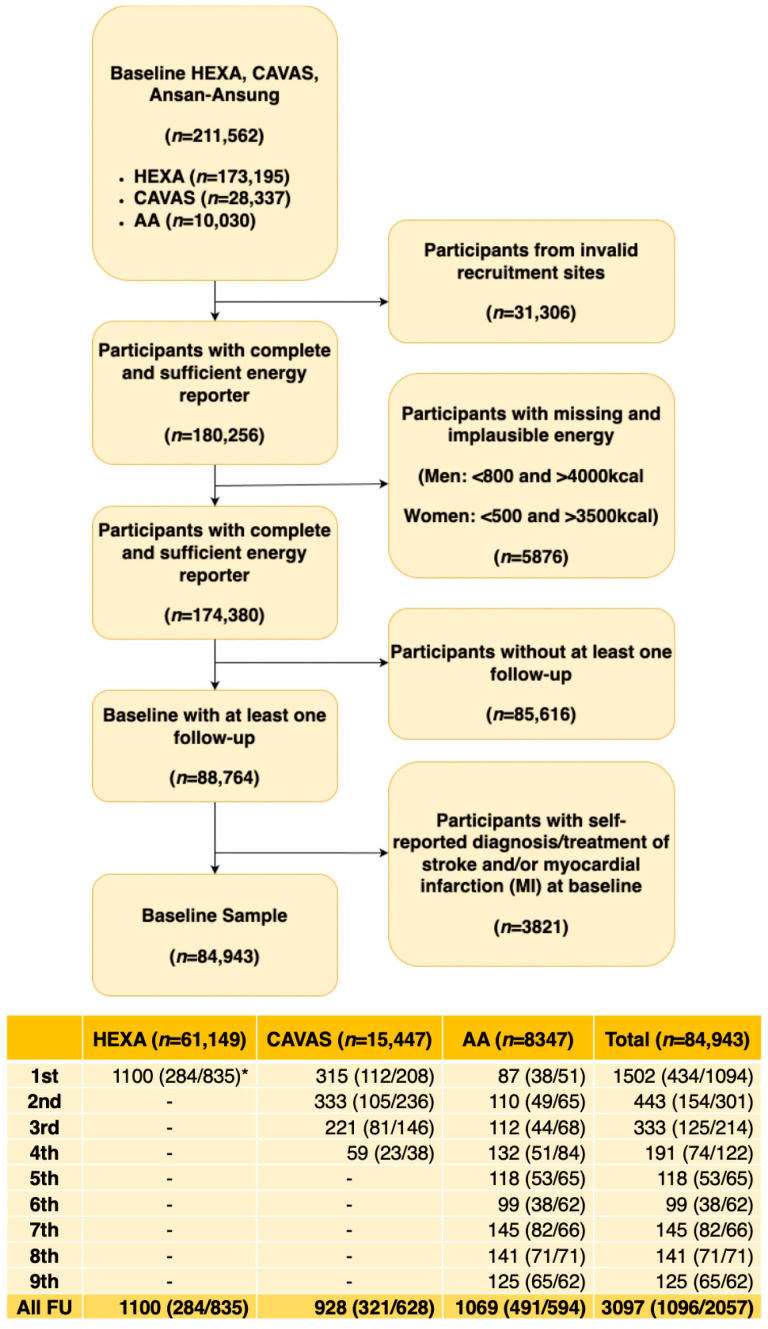
Selection criteria and identification of outcomes for all cohorts. * means reported numbers follow the format: Cases of either stroke or MI (Cases of stroke/Cases of MI).

**Figure 2 nutrients-17-01605-f002:**
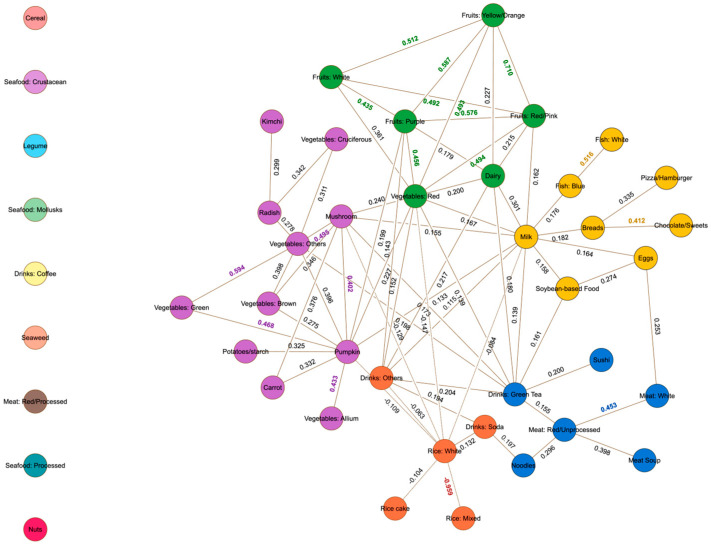
Food group network for all respondents derived using Gaussian graphical models. Nodes represent food groups. Edges represent conditional dependency between food groups measured by correlation coefficients. The absence of edges between two food groups represents conditional independence. Continuous lines represent positive conditional dependency, whereas broken lines represent negative conditional dependency. Nodes with the same color represent a community. Individual nodes outside the network do not belong to a community. Isolated nodes resulting from glasso regularization were not included in the community detection.

**Figure 3 nutrients-17-01605-f003:**
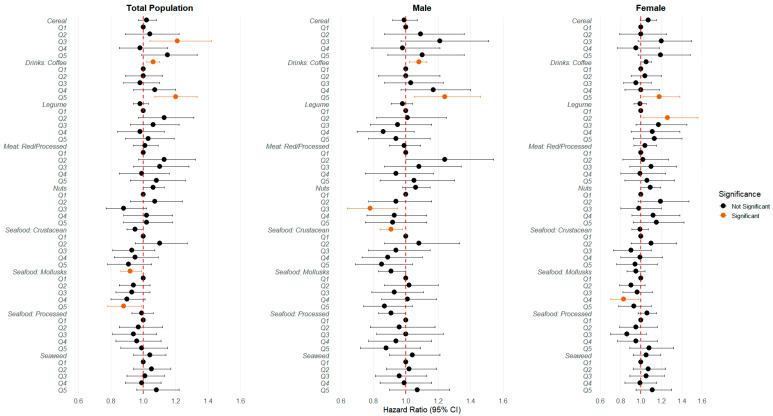
Sex-specific forest plot of hazard ratios for stroke or myocardial infarction across 9 food groups, adjusted for demographics and lifestyle.

**Table 1 nutrients-17-01605-t001:** Demographic variables and risk, event, and hazard ratio data of stroke or myocardial infarction for total population, males, and females.

Variable Category	Total Population (*n* = 84,729)			Male (*n* = 30,131)			Female (*n* = 54,598)		
	At Risk *n* (%)	No. of Events * *n* (%cases)	HR (95% CI)	At Risk *n* (%)	No. of Events * *n* (%cases)	HR (95% CI)	At Risk *n* (%)	No. of Events * *n* (%cases)	HR (95% CI)
Age: median (IQR)	54 (48–61)		1.08 (1.07–1.08)	55 (48–62)		1.06 (1.06–1.07)	53 (47–60)		1.09 (1.08–1.09)
Education level									
≤Middle school	34,407 (40.61)	1921 (62.03)	1.89 (1.70–2.09)	9489 (31.49)	673 (45.41)	1.60 (1.41–1.83)	24,918 (45.64)	1248 (77.28)	3.60 (2.91–4.46)
High school graduate	28,998 (34.22)	746 (24.09)	1.04 (0.92–1.17)	9946 (33.01)	470 (31.71)	1.17 (1.02–1.35)	19,052 (34.9)	276 (17.09)	1.34 (1.06–1.70)
College	21,324 (25.17)	430 (13.88)	1.00 (ref)	10,696 (35.5)	339 (22.87)	1.00 (ref)	10,628 (19.47)	91 (5.63)	1.00 (ref)
Income (million KRW)									
<1	11,269 (13.30)	816 (26.35)	2.11 (1.84–2.42)	3503 (11.63)	290 (19.57)	1.52 (1.26–1.84)	7766 (14.22)	526 (32.57)	3.26 (2.62–4.06)
1–1.99	15,387 (18.16)	631 (20.37)	1.47 (1.28–1.70)	5613 (18.63)	341 (23.01)	1.30 (1.08–1.56)	9774 (17.9)	290 (17.96)	1.85 (1.47–2.33)
2–3.99	33,014 (38.96)	786 (25.38)	1.10 (0.96–1.26)	11,816 (39.22)	440 (29.69)	1.02 (0.86–1.22)	21,198 (38.83)	346 (21.42)	1.30 (1.04–1.63)
≥4	15,533 (18.33)	274 (8.85)	1.00 (ref)	6070 (20.15)	177 (11.94)	1.00 (ref)	9463 (17.33)	97 (6.01)	1.00 (ref)
Unknown	9526 (11.24)	590 (19.05)	2.37 (2.05–2.74)	3129 (10.38)	234 (15.79)	1.92 (1.57–2.33)	6397 (11.72)	356 (22.04)	3.43 (2.73–4.30)
Size of the city									
Metropolitan city	39,578 (46.71)	726 (23.44)	1.00 (ref)	12,568 (41.71)	361 (24.36)	1.00 (ref)	27,010 (49.47)	365 (22.60)	1.00 (ref)
Middle-sized city	25,622 (30.24)	769 (24.83)	0.98 (0.88–1.09)	9873 (32.77)	445 (30.03)	0.85 (0.74–0.99)	15,749 (28.85)	324 (20.06)	0.98 (0.84–1.15)
Rural	19,529 (23.05)	1602 (51.73)	2.00 (1.82–2.19)	9946 (33.01)	676 (45.61)	1.34 (1.17–1.53)	11,839 (21.68)	926 (57.34)	2.63 (2.31–2.99)
Marriage status									
Married/living together	76,274 (90.02)	2732 (88.21)	1.00 (ref)	28,812 (95.62)	1422 (95.95)	1.00 (ref)	47,462 (86.93)	1310 (81.11)	1.00 (ref)
Single/unmarried	8455 (9.98)	365 (11.79)	1.37 (1.23–1.53)	1319 (4.38)	60 (4.05)	1.16 (0.90–1.50)	7136 (13.07)	305 (18.89)	1.66 (1.47–1.88)
Body mass index (BMI)									
<18.5	1443 (1.70)	50 (1.61)	1.45 (1.09–1.93)	434 (1.44)	29 (1.96)	1.45 (0.99–2.11)	1009 (1.85)	21 (1.3)	1.34 (0.86–2.07)
18.5 to 22.9	30,703 (36.24)	784 (25.31)	1.00 (ref)	8899 (29.53)	391 (26.38)	1.00 (ref)	21,804 (39.94)	393 (24.33)	1.00 (ref)
23.0 to 24.9	23,402 (27.62)	856 (27.64)	1.33 (1.21–1.47)	8871 (29.44)	415 (28.00)	1.13 (0.98–1.29)	14,531 (26.61)	441 (27.31)	1.48 (1.29–1.70)
≥25.0	29,181 (34.44)	1407 (45.43)	1.60 (1.46–1.74)	11,927 (39.58)	647 (43.66)	1.23 (1.09–1.40)	17,254 (31.60)	760 (47.06)	1.90 (1.68–2.14)
Waist circumference									
Normal range **	50,916 (60.09)	1505 (48.60)	1.00 (ref)	21,546 (71.51)	980 (66.13)	1.00 (ref)	29,370 (53.79)	525 (32.51)	1.00 (ref)
Above normal ***	33,813 (39.91)	1592 (51.40)	1.54 (1.44–1.65)	8585 (28.49)	502 (33.87)	1.45 (1.30–1.61)	25,228 (46.21)	1090 (67.49)	2.07 (1.86–2.29)
History of:									
Hypertension (Yes)	16,458 (19.42)	961 (31.03)	2.24 (2.07–2.41)	6559 (21.77)	417 (28.14)	1.86 (1.66–2.09)	9899 (18.13)	544 (33.68)	2.57 (2.32–2.85)
Hyperlipidemia (Yes)	7190 (8.49)	251 (8.10)	1.43 (1.25–1.62)	2564 (8.51)	113 (7.62)	1.26 (1.04–1.52)	4626 (8.47)	138 (8.54)	1.58 (1.32–1.88)
Diabetes mellitus (Yes)	5701 (6.73)	368 (11.88)	2.12 (1.90–2.36)	2699 (8.96)	193 (13.02)	1.82 (1.56–2.12)	3002 (5.50)	175 (10.84)	2.31 (1.98–2.70)
Job									
Professional or administrative	8516 (10.05)	158 (5.10)	1.00 (ref)	4649 (15.43)	137 (9.24)	1.00 (ref)	3867 (7.08)	21 (1.30)	1.00 (ref)
Office, sales, or service	16,565 (19.55)	320 (10.33)	1.06 (0.88–1.29)	7227 (23.99)	195 (13.16)	0.93 (0.75–1.16)	9338 (17.10)	125 (7.74)	2.42 (1.53–3.85)
Laborer or agricultural	20,875 (24.64)	1266 (40.88)	2.06 (1.74–2.43)	11,765 (39.05)	721 (48.65)	1.50 (1.25–1.80)	9110 (16.69)	545 (33.75)	5.84 (3.77–9.04)
Other, unemployed, or housekeeper	38,773 (45.76)	1353 (43.69)	1.32 (1.12–1.56)	6490 (21.54)	429 (28.95)	1.31 (1.08–1.59)	32,283 (59.13)	924 (57.21)	3.51 (2.27–5.41)
Smoking status									
Non-smoker	61,313 (72.36)	1862 (60.12)	1.00 (ref)	8428 (27.97)	338 (22.81)	1.00 (ref)	52,885 (96.86)	1524 (94.37)	1.00 (ref)
Past	12,901 (15.23)	596 (19.24)	1.49 (1.36–1.63)	12,258 (40.68)	570 (38.46)	1.18 (1.03–1.35)	643 (1.18)	26 (1.61)	1.53 (1.04–2.26)
Current	10,515 (12.41)	639 (20.63)	1.52 (1.39–1.67)	9445 (31.35)	574 (38.73)	1.20 (1.04–1.37)	1070 (1.96)	65 (4.02)	2.03 (1.59–2.61)
Drinking status									
Non-drinker	43,950 (51.87)	1539 (49.69)	1.000 (ref)	6181 (20.51)	325 (21.93)	1.00 (ref)	37,769 (69.18)	1214 (75.17)	1.00 (ref)
Past	3524 (4.16)	208 (6.72)	1.50 (1.30–1.73)	2426 (8.05)	154 (10.39)	1.15 (0.95–1.39)	1098 (2.01)	54 (3.34)	1.39 (1.06–1.82)
Current	37,255 (43.97)	1350 (43.59)	0.99 (0.92–1.06)	21,524 (71.43)	1003 (67.68)	0.86 (0.76–0.97)	15,731 (28.81)	347 (21.49)	0.72 (0.64–0.81)
Exercise									
Yes	41,473 (48.95)	1122 (36.23)	0.79 (0.74–0.85)	15,381 (51.05)	625 (42.17)	0.97 (0.87–1.08)	26,092 (47.79)	497 (30.77)	0.64 (0.58–0.71)

* event pertains to occurrence of stroke or myocardial infarction. ** normal range—Male: <90 cm; Female: <80 cm. *** above normal range—Male: ≥90 cm; Female: ≥80 cm.

**Table 2 nutrients-17-01605-t002:** Hazard ratios for networks scores and risk of stroke or myocardial infarction.

Network Scores: HR (95% CI)	Total Population (*n* = 84,729)				Males (*n* = 30,131)				Females (*n* = 54,598)				*p* for Interaction
	Cases	Model 1	Model 2	Model 3	Cases	Model 1	Model 2	Model 3	Cases	Model 1	Model 2	Model 3	
HCPF	3097	0.98 (0.97–0.99)	0.99 (0.98–1.00)	1.00 (0.98–1.01)	1482	0.99 (0.97–1.01)	0.99 (0.98–1.01)	0.99 (0.97–1.01)	1615	0.97 (0.96–0.99)	0.99 (0.98–1.01)	1.01 (0.99–1.02)	0.057
Q1	903	1.00 (ref)	1.00 (ref)	1.00 (ref)	386	1.00 (ref)	1.00 (ref)	1.00 (ref)	517	1.00 (ref)	1.00 (ref)	1.00 (ref)	
Q2	665	0.97 (0.88–1.07)	1.01 (0.91–1.12)	1.03 (0.93–1.14)	344	1.01 (0.87–1.16)	1.00 (0.86–1.16)	1.01 (0.87–1.17)	321	0.96 (0.83–1.10)	1.04 (0.90–1.20)	1.07 (0.93–1.23)	
Q3	582	0.92 (0.82–1.02)	0.97 (0.87–1.08)	0.99 (0.89–1.10)	273	0.92 (0.79–1.07)	0.92 (0.79–1.08)	0.93 (0.80–1.10)	309	0.93 (0.81–1.08)	1.04 (0.90–1.20)	1.07 (0.93–1.24)	
Q4	514	0.90 (0.80–1.00)	0.98 (0.87–1.09)	1.01 (0.90–1.13)	253	0.97 (0.83–1.14)	0.98 (0.84–1.15)	1.00 (0.84–1.18)	261	0.86 (0.74–1.00)	1.02 (0.87–1.19)	1.07 (0.91–1.25)	
Q5	433	0.82 (0.73–0.92)	0.92 (0.82–1.04)	0.96 (0.84–1.09)	226	0.94 (0.79–1.11)	0.95 (0.80–1.12)	0.96 (0.79–1.15)	207	0.75 (0.63–0.88)	0.95 (0.80–1.13)	1.02 (0.84–1.23)	
FD	3097	0.99 (0.99–1.00)	1.00 (0.99–1.01)	1.00 (1.00–1.01)	1482	1.00 (0.99–1.01)	1.00 (0.99–1.01)	1.00 (0.99–1.01)	1615	0.99 (0.98–1.00)	1.00 (0.99–1.01)	1.01 (1.00–1.02)	0.694
Q1	749	1.00 (ref)	1.00 (ref)	1.00 (ref)	415	1.00 (ref)	1.00 (ref)	1.00 (ref)	334	1.00 (ref)	1.00 (ref)	1.00 (ref)	
Q2	657	0.92 (0.83–1.03)	0.95 (0.85–1.05)	0.97 (0.87–1.08)	314	0.86 (0.74–1.00)	0.87 (0.75–1.00)	0.88 (0.76–1.02)	343	1.00 (0.86–1.16)	1.04 (0.89–1.21)	1.07 (0.92–1.25)	
Q3	609	0.94 (0.85–1.05)	0.99 (0.89–1.10)	1.02 (0.91–1.14)	300	0.93 (0.80–1.08)	0.95 (0.81–1.10)	0.96 (0.83–1.12)	309	0.97 (0.83–1.13)	1.05 (0.90–1.23)	1.09 (0.93–1.28)	
Q4	536	0.89 (0.80–1.00)	0.94 (0.84–1.05)	0.99 (0.88–1.11)	222	0.80 (0.68–0.95)	0.82 (0.69–0.96)	0.84 (0.71–0.99)	314	1.00 (0.85–1.16)	1.10 (0.94–1.28)	1.17 (1.00–1.38)	
Q5	546	0.94 (0.84–1.05)	1.01 (0.90–1.13)	1.08 (0.95–1.22)	231	1.01 (0.86–1.19)	1.03 (0.87–1.21)	1.06 (0.88–1.26)	315	0.93 (0.79–1.08)	1.05 (0.90–1.23)	1.16 (0.98–1.38)	
HPGT	3097	0.96 (0.94–0.98)	0.98 (0.96–0.99)	0.98 (0.96–1.00)	1482	0.98 (0.96–1.00)	0.98 (0.95–1.00)	0.97 (0.95–1.00)	1615	0.94 (0.92–0.97)	0.98 (0.96–1.01)	0.99 (0.96–1.02)	0.026
Q1	820	1.00 (ref)	1.00 (ref)	1.00 (ref)	221	1.00 (ref)	1.00 (ref)	1.00 (ref)	599	1.00 (ref)	1.00 (ref)	1.00 (ref)	
Q2	634	0.87 (0.78–0.96)	0.92 (0.83–1.02)	0.93 (0.83–1.03)	268	0.94 (0.79–1.12)	0.95 (0.80–1.14)	0.95 (0.80–1.14)	366	0.87 (0.76–0.99)	0.94 (0.82–1.08)	0.96 (0.84–1.10)	
Q3	551	0.79 (0.71–0.88)	0.85 (0.76–0.95)	0.86 (0.76–0.96)	294	0.94 (0.78–1.12)	0.92 (0.77–1.11)	0.92 (0.77–1.11)	257	0.73 (0.63–0.84)	0.84 (0.72–0.98)	0.86 (0.73–1.00)	
Q4	554	0.79 (0.71–0.89)	0.87 (0.77–0.98)	0.88 (0.78–1.00)	335	0.95 (0.80–1.13)	0.95 (0.79–1.13)	0.94 (0.78–1.13)	219	0.70 (0.60–0.83)	0.85 (0.72–1.01)	0.88 (0.74–1.05)	
Q5	538	0.77 (0.68–0.87)	0.86 (0.76–0.98)	0.88 (0.77–1.01)	364	0.87 (0.73–1.03)	0.86 (0.72–1.04)	0.85 (0.70–1.04)	174	0.73 (0.61–0.87)	0.94 (0.78–1.13)	0.99 (0.81–1.21)	
RHCB	3097	1.01 (0.99–1.02)	1.00 (0.99–1.02)	1.00 (0.99–1.02)	1482	1.00 (0.99–1.02)	1.01 (0.99–1.03)	1.01 (0.99–1.03)	1615	1.00 (0.98–1.02)	1.00 (0.98–1.01)	1.00 (0.98–1.02)	0.936
Q1	548	1.00 (ref)	1.00 (ref)	1.00 (ref)	232	1.00 (ref)	1.00 (ref)	1.00 (ref)	316	1.00 (ref)	1.00 (ref)	1.00 (ref)	
Q2	475	0.98 (0.87–1.11)	1.01 (0.90–1.15)	1.03 (0.91–1.17)	200	0.96 (0.80–1.16)	0.98 (0.81–1.18)	0.99 (0.82–1.20)	275	1.00 (0.85–1.18)	1.05 (0.90–1.24)	1.08 (0.92–1.27)	
Q3	491	1.05 (0.93–1.19)	1.10 (0.97–1.24)	1.11 (0.99–1.26)	203	1.03 (0.86–1.25)	1.07 (0.88–1.29)	1.08 (0.90–1.31)	288	1.07 (0.91–1.25)	1.14 (0.97–1.34)	1.15 (0.98–1.36)	
Q4	677	1.06 (0.95–1.19)	1.08 (0.96–1.21)	1.08 (0.96–1.21)	318	0.96 (0.81–1.13)	1.00 (0.84–1.18)	0.99 (0.84–1.18)	359	1.17 (1.00–1.36)	1.16 (1.00–1.36)	1.16 (0.99–1.35)	
Q5	906	1.04 (0.94–1.17)	1.05 (0.94–1.17)	1.05 (0.94–1.18)	529	1.00 (0.85–1.17)	1.04 (0.88–1.22)	1.04 (0.88–1.22)	377	1.05 (0.91–1.23)	1.02 (0.88–1.20)	1.05 (0.89–1.23)	
VEG	3097	0.98 (0.97–0.99)	0.99 (0.98–1.00)	0.99 (0.98–1.01)	1482	0.99 (0.97–1.00)	0.98 (0.97–1.00)	0.98 (0.96–1.00)	1615	0.98 (0.96–1.00)	1.00 (0.98–1.01)	1.01 (0.99–1.02)	0.484
Q1	796	1.00 (ref)	1.00 (ref)	1.00 (ref)	385	1.00 (ref)	1.00 (ref)	1.00 (ref)	411	1.00 (ref)	1.00 (ref)	1.00 (ref)	
Q2	715	0.99 (0.90–1.10)	1.02 (0.92–1.13)	1.03 (0.93–1.15)	325	0.94 (0.81–1.09)	0.94 (0.81–1.09)	0.94 (0.81–1.09)	390	1.05 (0.91–1.21)	1.10 (0.96–1.27)	1.15 (1.00–1.32)	
Q3	624	0.98 (0.88–1.08)	1.02 (0.91–1.13)	1.04 (0.93–1.16)	313	1.02 (0.88–1.19)	1.02 (0.88–1.19)	1.02 (0.87–1.19)	311	0.94 (0.81–1.10)	1.03 (0.89–1.20)	1.08 (0.92–1.26)	
Q4	514	0.86 (0.77–0.96)	0.92 (0.82–1.03)	0.94 (0.84–1.06)	251	0.89 (0.76–1.05)	0.89 (0.76–1.05)	0.88 (0.74–1.04)	263	0.85 (0.73–0.99)	0.97 (0.83–1.14)	1.03 (0.87–1.22)	
Q5	448	0.86 (0.77–0.97)	0.93 (0.82–1.05)	0.96 (0.84–1.09)	208	0.87 (0.74–1.04)	0.87 (0.73–1.04)	0.86 (0.71–1.04)	240	0.87 (0.74–1.02)	1.02 (0.86–1.21)	1.10 (0.92–1.32)	

HR: hazard ratio; CI: confidence interval; Model 1: adjusted for age and sex (for total respondents only); Model 2: adjusted for education, city, marriage status, income group, job group, previous history of hypertension, hyperlipidemia, or diabetes, BMI, and waist circumference, added to Model 1; Model 3: adjusted for drinking status, smoking status, exercise, and total energy intake, added to Model 2; HCPF (High-Calorie and Processed Food) includes breads, chocolate/sweets, eggs, blue fish, white fish, soybean-based food, milk, and pizza/hamburgers; FD (Fruits and Dairy) includes yellow/orange fruit, white fruit, purple fruit, red/pink fruit, red vegetables, and dairy; HPGT (High-Protein and Green Tea) includes white meat, sushi, meat soup, unprocessed red meat, noodles, and green tea drinks; RHCB (Rice and High-Calorie Beverages) includes soda drinks, other drinks, rice cake, white rice, and mixed rice; VEG (Vegetables) includes mushrooms, other vegetables, kimchi, carrots, brown vegetables, cruciferous vegetables, pumpkin, radishes, allium vegetables, green vegetables, and potatoes/starch; *p* for interaction was computed using Model 3 with interaction term of sex*community.

**Table 3 nutrients-17-01605-t003:** Hazard ratios for networks scores and risk of stroke.

Network Scores: HR (95% CI)	Total Population (*n* = 84,729)				Males (*n* = 30,131)				Females (*n* = 54,598)				*p* for Interaction
	Cases	Model 1	Model 2	Model 3	Cases	Model 1	Model 2	Model 3	Cases	Model 1	Model 2	Model 3	
HCPF	1040	0.97 (0.95–0.99)	0.99 (0.97–1.01)	0.99 (0.97–1.01)	512	0.98 (0.95–1.01)	0.99 (0.96–1.02)	0.99 (0.96–1.02)	528	0.96 (0.93–0.98)	0.99 (0.96–1.02)	0.99 (0.96–1.02)	0.437
Q1	325	1.00 (ref)	1.00 (ref)	1.00 (ref)	149	1.00 (ref)	1.00 (ref)	1.00 (ref)	176	1.00 (ref)	1.00 (ref)	1.00 (ref)	
Q2	224	0.91 (0.77–1.09)	0.97 (0.82–1.15)	0.98 (0.83–1.17)	120	0.93 (0.73–1.18)	0.95 (0.74–1.21)	0.95 (0.75–1.22)	104	0.90 (0.71–1.15)	0.99 (0.77–1.27)	1.01 (0.79–1.29)	
Q3	187	0.83 (0.69–0.99)	0.91 (0.75–1.09)	0.93 (0.77–1.11)	82	0.73 (0.56–0.96)	0.77 (0.58–1.01)	0.78 (0.59–1.03)	105	0.94 (0.73–1.20)	1.07 (0.83–1.37)	1.10 (0.85–1.41)	
Q4	161	0.79 (0.65–0.96)	0.90 (0.74–1.09)	0.92 (0.75–1.13)	82	0.84 (0.64–1.10)	0.89 (0.67–1.17)	0.90 (0.68–1.20)	79	0.76 (0.58–1.00)	0.92 (0.70–1.21)	0.95 (0.72–1.27)	
Q5	143	0.75 (0.62–0.92)	0.89 (0.72–1.10)	0.92 (0.73–1.15)	79	0.87 (0.66–1.14)	0.94 (0.71–1.25)	0.94 (0.69–1.29)	64	0.66 (0.49–0.88)	0.86 (0.64–1.17)	0.91 (0.65–1.27)	
FD	1040	0.99 (0.98–1.00)	1.00 (0.99–1.01)	1.00 (0.99–1.02)	512	0.99 (0.97–1.01)	0.99 (0.98–1.01)	1.00 (0.98–1.02)	528	0.99 (0.98–1.01)	1.01 (0.99–1.02)	1.01 (0.99–1.03)	0.617
Q1	244	1.00 (ref)	1.00 (ref)	1.00 (ref)	146	1.00 (ref)	1.00 (ref)	1.00 (ref)	98	1.00 (ref)	1.00 (ref)	1.00 (ref)	
Q2	238	1.00 (0.84–1.20)	1.04 (0.87–1.24)	1.06 (0.88–1.27)	118	0.91 (0.71–1.16)	0.93 (0.73–1.19)	0.94 (0.74–1.21)	120	1.13 (0.86–1.47)	1.19 (0.91–1.55)	1.22 (0.93–1.60)	
Q3	204	0.95 (0.79–1.14)	1.01 (0.83–1.22)	1.04 (0.86–1.26)	100	0.88 (0.68–1.13)	0.91 (0.70–1.17)	0.93 (0.71–1.21)	104	1.06 (0.80–1.39)	1.16 (0.88–1.53)	1.20 (0.90–1.59)	
Q4	186	0.93 (0.77–1.13)	1.01 (0.83–1.23)	1.06 (0.87–1.30)	75	0.76 (0.58–1.01)	0.81 (0.61–1.08)	0.84 (0.63–1.12)	111	1.14 (0.87–1.50)	1.28 (0.97–1.69)	1.36 (1.02–1.81)	
Q5	168	0.87 (0.71–1.06)	0.98 (0.80–1.20)	1.04 (0.83–1.29)	73	0.90 (0.68–1.19)	0.97 (0.73–1.29)	1.00 (0.73–1.36)	95	0.89 (0.67–1.19)	1.05 (0.79–1.41)	1.15 (0.84–1.58)	
HPGT	1040	0.95 (0.93–0.98)	0.98 (0.95–1.01)	0.98 (0.95–1.02)	512	0.98 (0.94–1.02)	0.99 (0.95–1.03)	0.99 (0.94–1.04)	528	0.93 (0.90–0.97)	0.98 (0.93–1.02)	0.98 (0.94–1.03)	0.420
Q1	284	1.00 (ref)	1.00 (ref)	1.00 (ref)	79	1.00 (ref)	1.00 (ref)	1.00 (ref)	205	1.00 (ref)	1.00 (ref)	1.00 (ref)	
Q2	190	0.73 (0.60–0.88)	0.79 (0.65–0.95)	0.80 (0.66–0.97)	83	0.82 (0.60–1.12)	0.87 (0.63–1.18)	0.87 (0.64–1.19)	107	0.70 (0.55–0.89)	0.77 (0.60–0.98)	0.78 (0.61–1.00)	
Q3	193	0.77 (0.64–0.93)	0.86 (0.71–1.04)	0.87 (0.72–1.06)	114	1.03 (0.77–1.39)	1.10 (0.82–1.47)	1.10 (0.81–1.49)	79	0.61 (0.47–0.80)	0.71 (0.54–0.94)	0.73 (0.56–0.97)	
Q4	186	0.73 (0.60–0.89)	0.85 (0.69–1.04)	0.87 (0.70–1.07)	118	0.96 (0.71–1.28)	1.05 (0.77–1.41)	1.05 (0.77–1.43)	68	0.59 (0.44–0.78)	0.72 (0.54–0.97)	0.75 (0.55–1.02)	
Q5	187	0.72 (0.59–0.89)	0.87 (0.70–1.07)	0.90 (0.71–1.14)	118	0.81 (0.60–1.10)	0.91 (0.66–1.25)	0.91 (0.64–1.28)	69	0.74 (0.55–0.99)	0.97 (0.71–1.32)	1.03 (0.74–1.45)	
RHCB	1040	1.01 (0.99–1.03)	1.00 (0.98–1.03)	1.00 (0.98–1.03)	512	1.03 (0.99–1.06)	1.02 (0.99–1.05)	1.02 (0.99–1.05)	528	0.99 (0.96–1.03)	0.99 (0.96–1.02)	0.99 (0.96–1.02)	0.593
Q1	188	1.00 (ref)	1.00 (ref)	1.00 (ref)	77	1.00 (ref)	1.00 (ref)	1.00 (ref)	111	1.00 (ref)	1.00 (ref)	1.00 (ref)	
Q2	138	0.84 (0.67–1.04)	0.89 (0.71–1.11)	0.90 (0.72–1.12)	64	0.94 (0.67–1.30)	0.97 (0.70–1.35)	0.99 (0.71–1.38)	74	0.77 (0.58–1.04)	0.83 (0.62–1.11)	0.84 (0.63–1.13)	
Q3	139	0.87 (0.70–1.08)	0.92 (0.74–1.15)	0.94 (0.75–1.17)	49	0.75 (0.53–1.08)	0.79 (0.55–1.13)	0.81 (0.56–1.15)	90	0.95 (0.72–1.26)	1.03 (0.78–1.36)	1.04 (0.79–1.38)	
Q4	227	1.00 (0.82–1.21)	0.99 (0.82–1.21)	0.99 (0.81–1.20)	114	1.02 (0.76–1.36)	1.02 (0.76–1.37)	1.01 (0.75–1.35)	113	0.99 (0.77–1.29)	0.98 (0.75–1.28)	0.98 (0.75–1.27)	
Q5	348	1.05 (0.87–1.26)	1.03 (0.86–1.24)	1.03 (0.85–1.25)	208	1.14 (0.87–1.49)	1.14 (0.86–1.50)	1.13 (0.86–1.49)	140	0.97 (0.75–1.25)	0.94 (0.72–1.22)	0.95 (0.73–1.24)	
VEG	1040	0.97 (0.95–0.99)	0.99 (0.97–1.01)	0.99 (0.97–1.01)	512	0.99 (0.96–1.02)	0.99 (0.97–1.02)	1.00 (0.96–1.03)	528	0.96 (0.93–0.99)	0.98 (0.95–1.01)	0.98 (0.95–1.01)	0.241
Q1	276	1.00 (ref)	1.00 (ref)	1.00 (ref)	128	1.00 (ref)	1.00 (ref)	1.00 (ref)	148	1.00 (ref)	1.00 (ref)	1.00 (ref)	
Q2	257	1.00 (0.85–1.19)	1.04 (0.87–1.23)	1.05 (0.89–1.25)	127	1.09 (0.85–1.40)	1.12 (0.87–1.43)	1.12 (0.87–1.44)	130	0.93 (0.73–1.18)	0.98 (0.77–1.25)	1.00 (0.79–1.28)	
Q3	202	0.89 (0.74–1.07)	0.94 (0.78–1.13)	0.96 (0.79–1.16)	107	1.05 (0.81–1.36)	1.08 (0.83–1.41)	1.09 (0.83–1.42)	95	0.76 (0.58–0.98)	0.83 (0.63–1.08)	0.85 (0.65–1.11)	
Q4	157	0.75 (0.61–0.91)	0.81 (0.66–1.00)	0.83 (0.67–1.02)	75	0.80 (0.60–1.07)	0.84 (0.62–1.12)	0.84 (0.62–1.13)	82	0.70 (0.53–0.92)	0.80 (0.61–1.06)	0.83 (0.62–1.11)	
Q5	148	0.82 (0.67–1.01)	0.92 (0.75–1.14)	0.95 (0.75–1.18)	75	0.97 (0.73–1.29)	1.03 (0.77–1.39)	1.03 (0.75–1.43)	73	0.71 (0.54–0.95)	0.85 (0.63–1.14)	0.88 (0.64–1.21)	

HR: hazard ratio; CI: confidence interval; Model 1: adjusted for age and sex (for total respondents only); Model 2: adjusted for education, city, marriage status, income group, job group, previous history of hypertension, hyperlipidemia, or diabetes, BMI, and waist circumference, added to Model 1; Model 3: adjusted for drinking status, smoking status, exercise, and total energy intake, added to Model 2; HCPF (High-Calorie and Processed Food) includes breads, chocolate/sweets, eggs, blue fish, white fish, soybean-based food, milk, and pizza/hamburgers; FD (Fruits and Dairy) includes yellow/orange fruit, white fruit, purple fruit, red/pink fruit, red vegetables, and dairy; HPGT (High-Protein and Green Tea) includes white meat, sushi, meat soup, unprocessed red meat, noodles, and green tea drinks; RHCB (Rice and High-Calorie Beverages) includes soda drinks, other drinks, rice cake, white rice, and mixed rice; VEG (Vegetables) includes mushrooms, other vegetables, kimchi, carrots, brown vegetables, cruciferous vegetables, pumpkin, radishes, allium vegetables, green vegetables, and potatoes/starch; *p* for interaction was computed using Model 3 with interaction term of sex*community.

**Table 4 nutrients-17-01605-t004:** Hazard ratios for networks scores and risk of myocardial infarction.

Network Scores: HR (95% CI)	Total Population (*n* = 84,729)				Males (*n* = 30,131)				Females (*n* = 54,598)				*p* for Interaction
	Cases	Model 1	Model 2	Model 3	Cases	Model 1	Model 2	Model 3	Cases	Model 1	Model 2	Model 3	
HCPF	2001	0.98 (0.97–1.00)	0.99 (0.98–1.01)	1.00 (0.98–1.02)	947	0.99 (0.98–1.02)	0.99 (0.97–1.02)	0.99 (0.97–1.02)	1054	0.98 (0.96–1.00)	1.00 (0.98–1.02)	1.01 (0.99–1.04)	0.072
Q1	555	1.00 (ref)	1.00 (ref)	1.00 (ref)	228	1.00 (ref)	1.00 (ref)	1.00 (ref)	327	1.00 (ref)	1.00 (ref)	1.00 (ref)	
Q2	433	1.02 (0.90–1.16)	1.05 (0.93–1.20)	1.07 (0.94–1.22)	221	1.08 (0.90–1.30)	1.06 (0.88–1.28)	1.07 (0.89–1.29)	212	1.00 (0.84–1.19)	1.08 (0.91–1.29)	1.12 (0.93–1.33)	
Q3	385	0.98 (0.86–1.11)	1.01 (0.89–1.16)	1.04 (0.91–1.19)	186	1.05 (0.86–1.27)	1.02 (0.84–1.24)	1.04 (0.85–1.27)	199	0.95 (0.79–1.13)	1.04 (0.87–1.25)	1.08 (0.90–1.29)	
Q4	346	0.97 (0.85–1.12)	1.04 (0.90–1.19)	1.07 (0.92–1.23)	169	1.09 (0.89–1.33)	1.06 (0.87–1.30)	1.07 (0.87–1.32)	177	0.92 (0.76–1.11)	1.08 (0.90–1.31)	1.14 (0.94–1.39)	
Q5	282	0.86 (0.75–1.00)	0.94 (0.81–1.10)	0.98 (0.83–1.16)	143	0.99 (0.80–1.22)	0.96 (0.78–1.20)	0.97 (0.76–1.23)	139	0.80 (0.65–0.99)	1.00 (0.81–1.24)	1.08 (0.85–1.36)	
FD	2001	1.00 (0.99–1.01)	1.00 (0.99–1.01)	1.00 (0.99–1.01)	947	1.00 (0.99–1.01)	1.00 (0.99–1.01)	1.00 (0.99–1.02)	1054	0.99 (0.98–1.01)	1.00 (0.99–1.02)	1.01 (1.00–1.02)	0.341
Q1	486	1.00 (ref)	1.00 (ref)	1.00 (ref)	259	1.00 (ref)	1.00 (ref)	1.00 (ref)	227	1.00 (ref)	1.00 (ref)	1.00 (ref)	
Q2	406	0.89 (0.78–1.02)	0.91 (0.80–1.04)	0.93 (0.81–1.06)	190	0.84 (0.69–1.01)	0.84 (0.69–1.01)	0.85 (0.70–1.03)	216	0.94 (0.78–1.14)	0.98 (0.81–1.18)	1.01 (0.84–1.22)	
Q3	401	0.97 (0.85–1.10)	1.00 (0.88–1.15)	1.04 (0.90–1.19)	197	0.99 (0.82–1.19)	0.99 (0.82–1.19)	1.00 (0.83–1.21)	204	0.96 (0.79–1.16)	1.04 (0.86–1.25)	1.08 (0.89–1.32)	
Q4	338	0.88 (0.76–1.01)	0.91 (0.79–1.05)	0.95 (0.82–1.10)	146	0.85 (0.69–1.04)	0.85 (0.69–1.04)	0.87 (0.70–1.07)	192	0.92 (0.76–1.11)	0.99 (0.82–1.21)	1.07 (0.88–1.31)	
Q5	370	0.99 (0.87–1.14)	1.04 (0.90–1.19)	1.11 (0.95–1.29)	155	1.09 (0.90–1.34)	1.07 (0.88–1.32)	1.11 (0.89–1.38)	215	0.96 (0.79–1.16)	1.06 (0.87–1.29)	1.18 (0.95–1.45)	
HPGT	2001	0.96 (0.95–0.99)	0.97 (0.95–1.00)	0.98 (0.95–1.00)	947	0.98 (0.95–1.01)	0.97 (0.94–1.00)	0.96 (0.93–1.00)	1054	0.95 (0.92–0.98)	0.99 (0.96–1.02)	0.99 (0.96–1.03)	0.033
Q1	521	1.00 (ref)	1.00 (ref)	1.00 (ref)	138	1.00 (ref)	1.00 (ref)	1.00 (ref)	383	1.00 (ref)	1.00 (ref)	1.00 (ref)	
Q2	427	0.93 (0.82–1.06)	0.97 (0.85–1.11)	0.98 (0.86–1.12)	180	1.01 (0.81–1.26)	1.00 (0.80–1.25)	0.99 (0.79–1.25)	247	0.94 (0.80–1.11)	1.01 (0.86–1.19)	1.03 (0.87–1.21)	
Q3	347	0.80 (0.69–0.91)	0.83 (0.72–0.96)	0.84 (0.73–0.97)	174	0.88 (0.70–1.10)	0.84 (0.66–1.05)	0.83 (0.66–1.05)	173	0.79 (0.66–0.95)	0.90 (0.75–1.08)	0.92 (0.76–1.12)	
Q4	360	0.83 (0.72–0.95)	0.88 (0.76–1.02)	0.89 (0.77–1.04)	213	0.96 (0.77–1.19)	0.90 (0.72–1.13)	0.89 (0.71–1.13)	147	0.77 (0.63–0.94)	0.92 (0.75–1.13)	0.95 (0.77–1.18)	
Q5	346	0.80 (0.69–0.93)	0.86 (0.74–1.01)	0.87 (0.74–1.04)	242	0.90 (0.72–1.13)	0.84 (0.67–1.06)	0.83 (0.64–1.06)	104	0.72 (0.57–0.90)	0.92 (0.72–1.16)	0.96 (0.74–1.24)	
RHCB	2001	1.00 (0.99–1.02)	1.00 (0.99–1.02)	1.00 (0.99–1.02)	947	1.00 (0.97–1.02)	1.00 (0.98–1.03)	1.00 (0.98–1.03)	1054	1.00 (0.98–1.03)	1.00 (0.98–1.02)	1.00 (0.98–1.03)	0.840
Q1	347	1.00 (ref)	1.00 (ref)	1.00 (ref)	151	1.00 (ref)	1.00 (ref)	1.00 (ref)	196	1.00 (ref)	1.00 (ref)	1.00 (ref)	
Q2	327	1.06 (0.91–1.23)	1.08 (0.93–1.26)	1.10 (0.95–1.28)	129	0.95 (0.75–1.20)	0.96 (0.76–1.21)	0.97 (0.77–1.23)	198	1.16 (0.95–1.42)	1.21 (0.99–1.47)	1.24 (1.02–1.51)	
Q3	344	1.16 (0.99–1.35)	1.20 (1.03–1.39)	1.22 (1.05–1.42)	151	1.18 (0.94–1.48)	1.20 (0.96–1.51)	1.22 (0.97–1.53)	193	1.15 (0.95–1.41)	1.22 (0.99–1.49)	1.23 (1.01–1.51)	
Q4	439	1.11 (0.96–1.28)	1.13 (0.98–1.31)	1.13 (0.98–1.31)	199	0.93 (0.75–1.15)	0.98 (0.79–1.22)	0.98 (0.79–1.22)	240	1.29 (1.07–1.56)	1.29 (1.06–1.56)	1.28 (1.06–1.55)	
Q5	544	1.05 (0.91–1.20)	1.05 (0.92–1.21)	1.07 (0.92–1.23)	317	0.94 (0.77–1.14)	1.00 (0.82–1.22)	1.00 (0.81–1.23)	227	1.10 (0.90–1.34)	1.06 (0.87–1.30)	1.09 (0.89–1.33)	
VEG	2001	0.99 (0.97–1.00)	0.99 (0.98–1.01)	0.99 (0.98–1.01)	947	0.98 (0.96–1.00)	0.98 (0.96–1.00)	0.98 (0.95–1.00)	1054	0.99 (0.97–1.01)	1.00 (0.99–1.02)	1.02 (0.99–1.04)	0.904
Q1	504	1.00 (ref)	1.00 (ref)	1.00 (ref)	250	1.00 (ref)	1.00 (ref)	1.00 (ref)	254	1.00 (ref)	1.00 (ref)	1.00 (ref)	
Q2	445	0.99 (0.87–1.12)	1.01 (0.88–1.14)	1.02 (0.90–1.16)	193	0.86 (0.72–1.04)	0.86 (0.71–1.04)	0.85 (0.70–1.03)	252	1.12 (0.94–1.33)	1.17 (0.98–1.40)	1.22 (1.02–1.46)	
Q3	413	1.03 (0.90–1.17)	1.07 (0.93–1.22)	1.09 (0.95–1.24)	201	1.01 (0.84–1.22)	1.00 (0.83–1.21)	0.98 (0.81–1.19)	212	1.07 (0.89–1.29)	1.16 (0.96–1.40)	1.23 (1.01–1.49)	
Q4	348	0.93 (0.81–1.07)	0.97 (0.85–1.12)	1.00 (0.86–1.16)	172	0.94 (0.78–1.15)	0.92 (0.75–1.12)	0.90 (0.73–1.11)	176	0.94 (0.77–1.14)	1.07 (0.87–1.30)	1.15 (0.93–1.41)	
Q5	291	0.88 (0.76–1.02)	0.93 (0.80–1.08)	0.96 (0.81–1.13)	131	0.84 (0.68–1.04)	0.81 (0.65–1.01)	0.79 (0.62–1.00)	160	0.95 (0.78–1.16)	1.10 (0.89–1.35)	1.21 (0.96–1.51)	

HR: hazard ratio; CI: confidence interval; Model 1: adjusted for age and sex (for total respondents only); Model 2: adjusted for education, city, marriage status, income group, job group, previous history of hypertension, hyperlipidemia, or diabetes, BMI, and waist circumference, added to Model 1; Model 3: adjusted for drinking status, smoking status, exercise, and total energy intake, added to Model 2; HCPF (High-Calorie and Processed Food) includes breads, chocolate/sweets, eggs, blue fish, white fish, soybean-based food, milk, and pizza/hamburgers; FD (Fruits and Dairy) includes yellow/orange fruit, white fruit, purple fruit, red/pink fruit, red vegetables, and dairy; HPGT (High-Protein and Green Tea) includes white meat, sushi, meat soup, unprocessed red meat, noodles, and green tea drinks; RHCB (Rice and High-Calorie Beverages) includes soda drinks, other drinks, rice cake, white rice, and mixed rice; VEG (Vegetables) includes mushrooms, other vegetables, kimchi, carrots, brown vegetables, cruciferous vegetables, pumpkin, radishes, allium vegetables, green vegetables, and potatoes/starch; *p* for interaction was computed using Model 3 with interaction term of sex*community.

## Data Availability

Data from Health Examinees (HEXA), Cardiovascular Disease Association Study (CAVAS), and Ansan-Ansung (AA) studies are part of the Korean Genome and Epidemiology Study (KoGES), managed by CODA (Clinical & Omics Data Archive), the Agency for Disease Control and Prevention, Republic of Korea. Due to legal restrictions on sharing sensitive patient information, the datasets used in this study cannot be made publicly available by the authors. Researchers seeking access to the data must provide ethics approval and a detailed research plan to CODA. Applications can be submitted through their website: https://coda.nih.go.kr/frt/index.do.

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
