# Peer review of "Network Approach to Evaluate the Effect of Diet on Stroke or Myocardial Infarction Using Gaussian Graphical Model"

_nutrients, 2025, doi:10.3390/nu17101605_

Round 1

Reviewer 1 Report

Comments and Suggestions for Authors

The manuscript presents an innovative approach by applying Gaussian Graphical Models (GGMs) to identify dietary patterns and their associations with stroke and myocardial infarction (MI) risk in a large Korean cohort. The paper leverages data from three well-established Korean cohorts and applies Cox proportional hazards models to examine associations. The study is methodologically ambitious and has potential implications for public health dietary recommendations in East Asian populations. However, several conceptual, methodological, and interpretative limitations undermine the robustness and reproducibility of the findings. Improvement suggestions:

  1. GGM assumes multivariate normality of the input data, which is rarely satisfied in dietary data characterized by zero inflation, skewness, and heavy tails. Although the authors applied a log transformation, the rationale behind the selected transformation (log10(1+g/day)) and the extent to which normality was achieved post-transformation are not evaluated. Furthermore, the implications of residual non-normality for the accuracy of inferred conditional dependencies and network sparsity should be discussed in detail.
  2. The method section mentions the use of the “huge” package in R for selecting the lambda parameter used in the graphical lasso algorithm. However, the manuscript fails to elaborate on the specific criterion applied for lambda selection. Given that the structure of the inferred network—and thus, the identification of dietary communities—is highly sensitive to this tuning parameter, it is necessary to justify and report the selection criterion transparently.
  3. The formation of five dietary communities based on network clustering is a central component of the analysis. However, the clustering algorithm applied is not clear.
  4. The manuscript mentions that nine food groups were not assigned to any dietary community. This omission raises methodological concerns. GGMs generate a full conditional dependency graph, and community detection should partition all nodes unless explicitly pruned. It is unclear whether these ungrouped food items were due to zero-degree nodes, post-hoc exclusion, or arbitrary thresholding of edge weights.
  5. The paper should report any validation studies of the questionnaire items used (e.g., sensitivity, specificity) against medical records or administrative claims data. If such validation is lacking, a quantitative bias analysis estimating the extent of misclassification bias should be considered.
  6. Although the authors controlled for a rich set of demographic, socioeconomic, and lifestyle covariates, the models do not appear to adjust for total energy intake, which is a critical confounder in dietary epidemiology. Differences in total caloric intake can distort the associations between food groups and disease outcomes, particularly when dietary patterns include both high- and low-calorie foods. The inclusion of energy intake in the adjusted models would control for isocaloric substitution and reduce the risk of confounding by over- or under-reporting. Additionally, nutrient-level confounders (e.g., sodium, fiber, fat) should be at least discussed.
  7. While history of hypertension, diabetes, and hyperlipidemia are adjusted in the models, the study does not control for medication use (e.g., statins, antihypertensives, antiplatelet agents), which can independently modify stroke and MI risk. Moreover, the analysis does not stratify by or exclude individuals with preclinical atherosclerosis, prior cardiovascular procedures, or high C-reactive protein levels. Inclusion of these covariates or sensitivity analyses excluding such individuals would provide more precise estimates of the dietary effects.
  8. The use of baseline dietary assessments and follow-up incidence of stroke/MI supports temporal causality. However, reverse causation remains a concern. Individuals may alter their diets due to pre-existing subclinical symptoms or after receiving preventive health advice. The authors should report whether any dietary reassessments were conducted during follow-up. If dietary intake was only measured once, this limitation should be acknowledged, and the potential attenuation of associations over time due to changes in diet should be discussed.
  9. Although the study conducts separate analyses for males and females, the manuscript does not examine interaction terms to formally test for effect modification by sex. Given the sex-specific biological and behavioral differences in cardiovascular risk and dietary habits, this is a critical omission. Including multiplicative interaction terms in the models or conducting likelihood ratio tests between sex-stratified and pooled models would enable the authors to make more robust claims about sex-specific effects.
  10. The manuscript briefly suggests that green tea and high-protein foods reduce cardiovascular risk, but does not delve into potential biological mechanisms. A more thorough discussion of anti-inflammatory, endothelial, metabolic, or gut microbiota-related pathways implicated in these dietary components would enhance the biomedical relevance of the findings.
  11. STROBE and STROBE-nut checklists should be reported.

Author Response

Thank you very much for taking the time to review our manuscript and provide valuable feedback. We have carefully addressed all the comments and suggestions, and our detailed responses are provided below. The corresponding revisions and corrections have been incorporated into the manuscript and are highlighted in red text in the re-submitted files for your convenience.

Comment 1: GGM assumes multivariate normality of the input data, which is rarely satisfied in dietary data characterized by zero inflation, skewness, and heavy tails. Although the authors applied a log transformation, the rationale behind the selected transformation (log10(1+g/day)) and the extent to which normality was achieved post-transformation are not evaluated. Furthermore, the implications of residual non-normality for the accuracy of inferred conditional dependencies and network sparsity should be discussed in detail.

Response 1: The rationale for the log10(1+g/day) transformation and the extent of normality improvement have been addressed. Limitations related to residual non-normality were acknowledged, with recommendations for future work. (Lines: 160-164; 461-466)

Comment 2: The method section mentions the use of the “huge” package in R for selecting the lambda parameter used in the graphical lasso algorithm. However, the manuscript fails to elaborate on the specific criterion applied for lambda selection. Given that the structure of the inferred network—and thus, the identification of dietary communities—is highly sensitive to this tuning parameter, it is necessary to justify and report the selection criterion transparently.

Response 2: We have now clarified in the Methods section that the Stability Approach to Regularization Selection (StARS) was used to select the optimal value of lambda, as it prioritizes network stability, which is critical in high-dimensional settings. (Lines: 171-175)

Comment 3: The formation of five dietary communities based on network clustering is a central component of the analysis. However, the clustering algorithm applied is not clear.

Response 3: Thank you for pointing this out. We have now clarified that community detection was performed using Multi-Level Modularity Optimization implemented in the R package "igraph," which has been shown to perform best among igraph clustering algorithms in previous studies. (Lines: 178-181)

Comment 4: The manuscript mentions that nine food groups were not assigned to any dietary community. This omission raises methodological concerns. GGMs generate a full conditional dependency graph, and community detection should partition all nodes unless explicitly pruned. It is unclear whether these ungrouped food items were due to zero-degree nodes, post-hoc exclusion, or arbitrary thresholding of edge weights.

Response 4: We appreciate your careful attention to this point. The nine food groups were not assigned to any community because, after applying glasso regularization (a step to parse the network), they appeared as isolated nodes with no strong partial correlations. As Multi-Level Modularity Optimization (community detection) operates on connected components, these unlinked nodes were naturally excluded from the community partitioning, reflecting the intended sparsity of the model rather an exclusion. We have included additional figure caption to address this concern in Figure 2. (Lines: 246-247; 250-251)

Comment 5: The paper should report any validation studies of the questionnaire items used (e.g., sensitivity, specificity) against medical records or administrative claims data. If such validation is lacking, a quantitative bias analysis estimating the extent of misclassification bias should be considered.

Response 5: We have added information on the validation and reproducibility of the FFQ used in our study. The FFQ was validated against 12-day dietary records, with correlation coefficients ranging from 0.23 to 0.64, and showed reasonable reproducibility over a one-year interval, with correlation coefficients between 0.24 and 0.58. (Lines: 128-132)

Comment 6: Although the authors controlled for a rich set of demographic, socioeconomic, and lifestyle covariates, the models do not appear to adjust for total energy intake, which is a critical confounder in dietary epidemiology. Differences in total caloric intake can distort the associations between food groups and disease outcomes, particularly when dietary patterns include both high- and low-calorie foods. The inclusion of energy intake in the adjusted models would control for isocaloric substitution and reduce the risk of confounding by over- or under-reporting. Additionally, nutrient-level confounders (e.g., sodium, fiber, fat) should be at least discussed.

Response 6: We thank the reviewer for highlighting this critical issue. We reanalyzed our Model 3 by additionally adjusting for total energy intake. After this adjustment, most changes in the hazard ratios (HRs) were minor, typically around 0.01 to 0.02, indicating that the overall associations remained robust. We also corrected the citations in the Results section, Figure 3, and Supplementary Tables 4,5,6 with Model 3 changes accordingly, although these revisions did not substantially alter the interpretation of our findings. Regarding nutrient-level confounders such as sodium, fiber, and fat, we acknowledge their potential relevance. However, we decided not to adjust for individual nutrients in our primary models to avoid overadjustment. (Table 2,3,4; Supplementary Table 4,5,6)

Comment 7: While history of hypertension, diabetes, and hyperlipidemia are adjusted in the models, the study does not control for medication use (e.g., statins, antihypertensives, antiplatelet agents), which can independently modify stroke and MI risk. Moreover, the analysis does not stratify by or exclude individuals with preclinical atherosclerosis, prior cardiovascular procedures, or high C-reactive protein levels. Inclusion of these covariates or sensitivity analyses excluding such individuals would provide more precise estimates of the dietary effects.

Response 7: We appreciate this thoughtful comment. While data on medication use, preclinical atherosclerosis, and prior cardiovascular procedures were collected, these variables had high rates of missingness (>50%), particularly for medication use, which was only recorded during the later phases of data collection. Furthermore, the recorded procedures were largely unrelated to cardiovascular outcomes. Due to the limitations in data completeness, we were unable to include these variables in our main analyses. We agree that their inclusion or stratification could refine effect estimates and have added this as a limitation in the revised manuscript. (Lines: 453–460)

Comment 8: The use of baseline dietary assessments and follow-up incidence of stroke/MI supports temporal causality. However, reverse causation remains a concern. Individuals may alter their diets due to pre-existing subclinical symptoms or after receiving preventive health advice. The authors should report whether any dietary reassessments were conducted during follow-up. If dietary intake was only measured once, this limitation should be acknowledged, and the potential attenuation of associations over time due to changes in diet should be discussed.

Response 8: The limitation of baseline-only dietary measurement and follow-up assessments was acknowledged in the manuscript. We recognize that reverse causation and potential attenuation of associations due to dietary changes over time may affect the observed relationship between diet and stroke/MI incidence.  (Lines: 446-448)

Comment 9: Although the study conducts separate analyses for males and females, the manuscript does not examine interaction terms to formally test for effect modification by sex. Given the sex-specific biological and behavioral differences in cardiovascular risk and dietary habits, this is a critical omission. Including multiplicative interaction terms in the models or conducting likelihood ratio tests between sex-stratified and pooled models would enable the authors to make more robust claims about sex-specific effects.  

Response 9: We have included the interaction analysis on Tables 2,3,4 and Supplementary Tables 4,5,6 (Lines: 203-205; 254-255; 311-312; 351-353)

Comment 10: The manuscript briefly suggests that green tea and high-protein foods reduce cardiovascular risk, but does not delve into potential biological mechanisms. A more thorough discussion of anti-inflammatory, endothelial, metabolic, or gut microbiota-related pathways implicated in these dietary components would enhance the biomedical relevance of the findings.

Response 10: We have expanded the Discussion section to include relevant biological mechanisms underlying the effects of green tea and high-protein foods. Specifically, we discuss how catechins in green tea reduce oxidative stress and inflammation, promote endothelial function through nitric oxide production, and improve lipid and glucose metabolism—mechanisms that collectively reduce cardiovascular risk. (Lines: 332–336)

Comment 11: STROBE and STROBE-nut checklists should be reported.

Response 11: We have now included both checklist in the Supplementary Files

Reviewer 2 Report

Comments and Suggestions for Authors

Your study provides valuable insights into the dietary patterns affecting cardiovascular outcomes in the Korean population and employs a novel analytical approach using Gaussian Graphical Models.

The manuscript is well-structured, clearly written, and addresses an important public health topic. The methodology is robust, utilizing a substantial cohort with careful consideration of confounding variables. The innovative application of GGM significantly enhances our understanding of dietary interactions related to cardiovascular risk.

I would recommend some minor revisions to improve the clarity and strength of your paper:

- Discuss the advantages and assumptions underlying the GGM approach in comparison to other dietary pattern analyses.

- Clearly acknowledge potential limitations related to self-reported dietary and health outcome data, such as recall bias or misclassification, and discuss their implications.

- Streamline the presentation of your results, potentially summarizing extensive data tables and emphasizing the most significant findings.

- Provide brief explanations or hypotheses for the observed gender differences in dietary effects on cardiovascular outcomes to add depth to your conclusions.

Author Response

Thank you very much for taking the time to review this manuscript. Please find the detailed responses below and the corresponding revisions and corrections highlighted in red texts in the re-submitted files.

Comment 1: Discuss the advantages and assumptions underlying the GGM approach in comparison to other dietary pattern analyses.

Response 1: Thank you for pointing this out. We have rephrased and expanded the advantages of the Gaussian Graphical Model (GGM) in comparison to other dietary pattern analysis methods in the introduction. Additionally, we have addressed the assumptions underlying the GGM approach in the methods section. (Lines: 69-75; 148-149)

Comment 2: Clearly acknowledge potential limitations related to self-reported dietary and health outcome data, such as recall bias or misclassification, and discuss their implications.

Response 2: We have now acknowledged the potential for misclassification bias due to inaccuracies in self-reported disease history, as indicated by the previously reported positive predictive values for stroke and myocardial infarction (Lines: 440-442)

Comment 3: Streamline the presentation of your results, potentially summarizing extensive data tables and emphasizing the most significant findings.

Response 3: Thank you for your feedback. In the manuscript, we have summarized our key observations in the main text, condensing the extensive data tables to highlight the most significant findings. For readers who are interested in more detailed data, we have provided the full tables in the supplementary files, with references to these data included in the manuscript. This approach allows us to present the most relevant information concisely, while still making the complete dataset available for further examination.

Comment 4: Provide brief explanations or hypotheses for the observed gender differences in dietary effects on cardiovascular outcomes to add depth to your conclusions. 

Response 4: Thank you for the comment. We have already provided explanations for the observed gender differences in dietary effects on cardiovascular outcomes wherever significant results were noted in the discussion (Lines: 361-365; 373-376). However, we added additional brief explanation for the gender differences in the HPGT and VEG community. (Lines: 351-353; 379-380)

Round 2

Reviewer 1 Report

Comments and Suggestions for Authors

The authors succeeded in responding to all my previous concerns and convinced me to accept the manuscript in its present form.